# Microstructure and Mechanical Properties of Laser-Welded Joints between DP590 Dual-Phase Steel and 304 Stainless Steel with Preset Nickel Coating

**DOI:** 10.3390/ma16072774

**Published:** 2023-03-30

**Authors:** Hua Zhang, Jiahui Xu, Desheng Hao, Othman Mohammed Ali Othman Esmail

**Affiliations:** School of Mechanical Engineering, Nantong University, Nantong 226019, China

**Keywords:** laser welding, dissimilar joints, austenitic steel, dual phase steel, nickel coating, microstructure, mechanical properties

## Abstract

Dissimilarities in metal laser welding lead to brittleness in welded joints due to differences in the thermophysical and chemical properties between dissimilar base materials. To overcome such brittleness, the addition of a preset coating onto the base materials as an interlayer is a method for attaining reliable welded joints. Nd:YAG laser butt welding of DP590 dual-phase steel and 304 stainless, both with a thickness of 1 mm, was performed with a preset nickel coating as an interlayer using an electroplating process. The relationship between the microstructure and the mechanical properties of the welded joints was researched, the microstructure and composition of the weldment were analyzed, and the microhardness, tensile strength and corrosion resistance were tested. The results showed that the preset nickel coating increased the content of Ni element in the welded joints, which is beneficial to the formation of lath martensite. The average hardness of the welded joints increased by 12%, and the tensile strength was higher than 370 MPa. The corrosion rate of the welded joints can be slowed down, and the corrosion resistance can be improved by increasing the nickel coating.

## 1. Introduction

With the rapid development of the manufacturing industry, the performance of individual materials is only with difficulty able to meet the production needs of the diverse functions of mechanical products. Joints between dissimilar materials are an effective way of achieving this, and the fabrication of joints using dissimilar materials has been applied in various fields, including in the automobile, power generation, chemical, petrochemical, nuclear and electronics industries [1,2,3,4,5]. However, there are differences in thermophysical and chemical properties between dissimilar materials that may lead to the formation of brittle intermetallic phases in the weldment [6]. To overcome such brittleness, various filler materials have been used between dissimilar materials to suppress the formation of brittle intermetallic compounds. The forms of the filler materials include powder [7,8], filler wire [9,10] and interlayer [11,12,13].

Compared to powder and filler wire, the metal distribution added using interlayer is uniform. Xie et al. [14] investigated the microstructural and mechanical properties of the ultrasonic spot welding of the dissimilar materials TiNi/Ti6Al4V using a pure Al coating. The study revealed that the dissimilar materials TiNi/Ti6Al4V had been successfully joined and the tensile shear load reached a maximum value of 931 N. Varbai et al. [15] used gas tungsten arc welding to obtain a joint between lean duplex stainless steel and 304 austenitic stainless steel. Their investigations revealed that the best pitting corrosion resistance was achieved using a 308L welding rod with argon shielding gas, with a weight loss of 1.6% after the 24 h immersion test. Peng et al. [16] performed dissimilar laser overlap welding of 316L stainless steel to 6061 Al alloy using a Cu coating. The study revealed that the shear strength of the joints increased, while the formation of intermetallic compounds decreased with increasing Cu coating thickness. Wang et al. [17] performed electron beam welding of Ti-15-3 titanium alloy with 304 stainless steel both with the provision of an interlayer of copper sheet and without interlayer. The study suggested that the addition of copper element into the weld improved the metallurgical conditions, optimizing the plasticity and toughness of the joint. B. Shanmugarajan et al. [18] performed laser welding investigations on dissimilar Ti-SS using interlayers of Vanadium and Tantalum in the form of laser cladding. The investigations revealed that the use of Vanadium as interlayer was not able to achieve a crack-free joint, while a joint with a strength of 40 MPa could be formed using Tantalum as interlayer. Tomashchuk et al. [19] studied electron beam welding between titanium alloy and AISI 316L austenitic stainless steel using a copper foil as an intermediate layer. The study showed that the interface between the titanium alloy and the melted zone contained compact layers of Cu-Fe-Ti intermetallic and was more stable than the interface between the melted zone and steel. Hailat et al. [20] performed laser micro-welding of aluminum and copper with and without tin foil alloy as interlayer. The study revealed that the incorporation of tin foil alloy as filler was useful for improving the mechanical properties of the laser joint. Lee et al. [21] investigated the effect of using a copper insert layer on the properties of friction-welded joints between TiAl and AISI 4140. The study showed that pure copper could be used as an insert metal for a stress relief buffer layer in order to prevent crack formation at the interface. Mitelea et al. [22] performed laser welding of the dissimilar materials Ti-6Al-4V alloy and X5CrNi18-10 steel by inserting copper foil as interlayer. The study revealed that the ultimate tensile stress of the tested joints was lower than that of the base materials, but higher than that of the Cu intermediate layer. Compared to foil interlayer, electroplating is an economical and easy way of obtaining an interlayer. Chatterjee et al. [23] performed laser welding of the dissimilar materials Ti6Al4V and AISI 304 by means of copper deposition. The study revealed that the presence of copper at the welding edges was able to reduce brittleness. In addition, Huaibo Deng et al. [24] compared back-heating-assisted friction stir welding with conventional friction stir welding to confirm the feasibility of dissimilar NiTi/Ti6Al4V joints fabricated via friction stir welding. They found that a defect-free joint could be obtained when using a preheating temperature of 200 °C during back-heating-assisted friction stir welding. Chen Yuhua et al. [25] investigated the welding crack in micro-laser-welded joints between the dissimilar metals NiTiNb shape memory alloy and Ti6Al4V alloy. The study revealed that a higher content of the brittle Ti2Ni phase was the main reason for crack formation, along with a high degree of stress concentration.

Copper is generally adopted as an interlayer metal to reduce the amount of brittle intermetallic compound in dissimilar welding. To increase the corrosion resistance of the dissimilar weld, the use of nickel as the interlayer metal is more suitable. In this paper, DP590 steel and 304 stainless steel are laser welded with a preset nickel coating. Through a comparative analysis of the microstructure, hardness, tensile strength and corrosion resistance of the welded joints, the relationship between the microstructure and the mechanical properties is systematically studied.

## 2. Materials and Methods

### 2.1. Materials and Preparation

Laser welding of dissimilar materials was performed on DP590 dual-phase steel and 304 stainless steel plates with dimensions of 100 mm × 100 mm × 1 mm. The main chemical components of DP590 and 304 are shown in Table 1. Electroplating of nickel was carried out over the workpiece to provide nickel as interlayer. The workpiece (i.e., DP590 or 304) was connected to the cathode, while a nickel foil was connected to the anode. An electrolyte bath was prepared with 30 wt% (weight percentage) NiSO_4_, 4 wt% NiCl_2_, 4 wt% H_3_BO_3_, 0.03 wt% C12H25SO4Na and the rest distilled water. The current density was 4 Adm^−2^ and the electroplating time was 2 h. Before electroplating, the workpieces were cleaned with acetone to remove oil stains and weak acid to remove oxide.

### 2.2. Welding Process and Test Work

Dissimilar welding was carried out using a pulsed Nd:YAG laser welding machine (JHM-1GXY-500, CHUTIAN, Wuhan China). A schematic diagram of dissimilar laser welding is shown in Figure 1. The vertical incidence mode was adopted for the laser and the shielding gas supply port and the laser beam were at 30°. The laser was used in the process with 15–19 J single-pulse energy, a pulse width of 4 ms, and a frequency of 10 Hz. The welding speed was 150 mm/min. The shielding gas was argon, with a purity of 99.9% and a flow rate of 25 L/min.

In order to investigate the effect of a nickel coating on dissimilar welding, four samples of dissimilar laser welding were performed with different experimental conditions. The experimental conditions are shown in Table 2.

After the dissimilar laser welding had been performed, the welded workpiece was cut along the cross-section of the weld to obtain specimens. These specimens were made into metallographic samples to test the hardness and corrosion resistance of the welds with a microhardness tester (TMVS-1, Beijing Times, Beijing, China) and an electrochemical workstation (CHI666C, Shanghai Chenhua, Shanghai, China). The electrochemical workstation was a three-electrode system: the working electrode was the welded joint, the reference electrode was a saturated calomel electrode, and the auxiliary electrode was a platinum electrode. The working condition of the three-electrode system was 0.5% NaCl solution. To test the microstructure of the samples, the side of the weld close to the 304 stainless steel was etched in aqua regia, and the other side of the weld close to the DP590 steel was etched in 4% nitric acid. Microstructure observations were carried out on the samples using a confocal microscope (Axio Scope A1, ZEISS, Jena, Germany) and a scanning electron microscope (S-3400N, Hitachi, Japan). In order to test the tensile properties of the weld, the welded workpiece was cut into multiple tensile samples. The dimensions of the tensile sample are shown in Figure 2. The tensile sample obtained via dissimilar laser welding of DP590-304 with preset nickel coating is shown in Figure 3. An electronic universal tensile testing machine (CMT5105, MTS, Eden Prairie, MI, UUSA) was used to measure the tensile strength of the samples.

## 3. Results and Discussion

### 3.1. Microstructure

To compare dissimilar laser welding with and without the preset nickel coating, A welded sample (No. 1) was produced under the conditions shown in Table 2. The microstructures of the welded joint without the nickel coating are shown in Figure 4. With a single-pulse laser energy of 15 J, the welded joint was not penetrated. Figure 4a,b show the microstructure of the boundary zone near DP590 and 304, respectively. At the boundary zone near the DP590, the microstructure consists of martensites and a small number of ferrites. However, at the boundary zone near the 304, the microstructure consists only of martensite. The reason for this difference is that the thermal conductivity of DP590 is better than that of 304. During the dissimilar laser welding process, the weld pool near DP590 generates a thermal cycle peak temperature, forming a heat-affected zone (HAZ). The thermal cycle peak temperature leads to a decrease in cooling rate and an increase in the high-temperature residence time, which promotes the formation of ferrite.

Three samples (No. 2 to No. 4) were produced by performing dissimilar laser welding with the preset nickel coating at different single-pulse laser energies (15 J, 17 J, 19 J). The pool profiles of the three welded samples are showed in Figure 5a, Figure 6a and Figure 7a. From the pool profiles of the three samples, it can be seen that the welds with single-pulse energies of 15 J and 17 J exhibit no penetration, while the welded joint is penetrated under a single-pulse energy of 19 J. Comparing the profiles of the three welded samples, they have the same characteristics, i.e., a clear fusion line between the 304 base metal and the weld, but no clear fusion line between the DP590 base metal and the weld. As can be observed from the weld pools shown in Figure 5a, Figure 6a and Figure 7a, the weld pools are divided into two parts, with b and c indicating two regions in the weld zone, i.e., the microstructures of the intermetallic zone near 304 (region b) and the intermetallic zone near DP590 (region c). As shown in Figure 5b, Figure 6b and Figure 7b, region b in the three samples is characterized by the microstructure mainly consisting of martensites. As shown in Figure 5c, Figure 6c and Figure 7c, region c of three samples is characterized by the microstructures consisting not only of martensites, but also ferrites. It can be seen that the change law regarding the microstructures of the welded joints with the preset nickel coating is consistent with the welded joints without the nickel coating.

Figure 8 shows the microstructure of the fusion zone of the DP590-304 welded joints (where Figure 8a–d correspond to samples number 1–4, respectively). The microstructure of the fusion zone of the DP590-304 welded joints without the preset nickel coating is shown in Figure 6a. Based on the qualitative analysis, the basis for judging of austenite (A) is the white dot, while the basis for judging of lath martensite (LM) is the narrow profile. The structures in this area consist of lath martensite and a small amount of austenite. The microstructure of the fusion zone of the DP590-304 welded joints with the preset nickel coating under different single-pulse laser energies are shown in Figure 8b–d. Comparing the three SEM pictures to perform a qualitative analysis, it can be seen that the structures of the dissimilar welds under different single-pulse laser energies are characterized by the presence of less austenite and more lath martensite. The reason for this characteristic is that the increase in Ni content in the welded joints inhibits the formation of austenite and transforms the weld structure into lath martensite.

It is also found that the grain of the weld changed from dendritic crystal to cellular crystal, and with the increasement of single laser pulse energy, the cellular crystals at the weld are more uniform and meticulous.

### 3.2. Microhardness

Microhardness testing was performed on the lateral cross-section of the welded joint. The tested areas included base metal (DP590), weld pool and base metal (304). In the hardness test, a diamond-shaped indenter with a load of 0.245 N was used, and a dwell time of 15 s was considered. The comparative results for the hardness values of sample numbers 1–4 are plotted in Figure 9. The hardness values were measured from the center of the weld pool towards both sides of the dissimilar base materials (DP590 on the left and 304 on the right). From Figure 9, it can be clearly observed that the distribution of hardness of these four welded joints has the same trend, namely, that the microhardness value decreases from the center of the weld pool towards the base materials. The average hardness values of the welded joints of sample numbers 1–4 are 384.7 HV, 401.3 HV, 426.5 HV and 437 HV, respectively. The preset nickel coating improves the microhardness of the weld zone for dissimilar laser welding of DP590-304. The average hardness of Sample 4 is increased by 12% compared with Sample 1. This conclusion is consistent with the previous comparison of microstructures of sample numbers 1–4. The microstructure of the welded joints without the preset nickel coating consists of lath martensite and a small amount of austenite, while the microstructure of the welded joints with the preset nickel coating consists only of lath martensite.

Moreover, the energy spectrum of Ni element obtained by line scanning the weld interface of sample numbers 1–4 are shown in Figure 10. The ordinate in Figure 10 is dimensionless, which indicates the relative amount of nickel in the welded area. To compare the results of these four samples, an ellipse is used to indicate the level of nickel element near the side of the DP590 base metal. From Figure 10a, it is obvious that content of nickel element near DP590 is the lowest. The reason for this is that the DP590 base material does not contain Ni element, while the Ni element in the 304 steel flows to the DP590 side with the weld pool during laser welding. On the basis of a comparison of Figure 10b–d, it can be seen that the content of Ni element in the welded joints increased with increasing pulse laser energy. This means that more of the nickel coating melts into the molten pool, increasing the content of Ni in the welded joints. Combined with the analysis presented in Figure 9 and Figure 10, it can be found that the preset nickel coating increases the microhardness of the weld through the addition of nickel element.

Based on the Gaussian distribution characteristics of the laser heat source, the two-dimensional temperature fields of DP590/304 weld with the nickel coating under different pulse laser energies are shown in Figure 11. The temperature fields on both sides of the dissimilar steel weld are asymmetrically distributed, and the temperature gradient of the DP590 dual-phase steel is greater than that of the 304 stainless steel. This phenomenon is mainly due to the heat conduction speed of DP590 dual-phase steel being faster than that of 304 stainless steel. With increasing pulse laser energy, the temperature of the welded zone increases. Therefore, the temperature in the molten pool area increases, and the fluidity of the nickel element of the coating layer improves. This further illustrates the conclusions presented in Figure 9 and Figure 10.

### 3.3. Tensile Strength

Figure 12 shows the stress–displacement curves of Samples 1–4, which were drawn with the fracture displacement of the welded joints as the abscissa and the tensile strength of the welded joints as the ordinate. Tensile test results of the welded joints of the four samples are listed in Table 3. The tensile test data results of the welded joints from Samples 1–3 were obtained while fracturing at the welded joints, while Sample 4 was fractured on the side of the DP590 base material. Compared with the stress–displacement curves of sample numbers 1 (without the preset nickel coating) and 2 (with the preset nickel coating) under the same laser welding parameters, it is indicated that the preset nickel coating improves the tensile strength of the weld. The main reason for this is that the welded joint structure of Sample 1 is composed of lath martensite and a small amount of austenite in a disorderly distribution, while that of Sample 2 is composed of evenly distributed lath martensite. The maximum tensile strength of 370 MPa for the welded joint obtained for Sample 4 is better than that of Samples 2 and 3. The main reason for this is that the welds of Samples 2 and 3 exhibit no penetration, while the welded joint of Sample 4 is penetrated. Another reason is that the distribution of lath martensite in the weld area becomes more uniform with increasing single-pulse energy.

Figure 13 shows tensile fracture in Sample 4. The fracture zone is on the side of the DP590 base metal, and the distance between the fracture zone and the weld zone is about 8 mm. It can be seen that the deformation area in the tension of the welded sample is concentrated on the DP590 side, and there is no obvious deformation in the weld zone or in the 304 base metal. This also confirms that the tensile strength of the weld of Sample 4 is better than that of the DP590 base metal.

### 3.4. Corrosion Resistance

The corrosion resistance of the welds was tested using the electrochemical workstation. On the basis of the data obtained using the electrochemical workstation, the polarization curves of the four samples weld are shown in Figure 14. The four polarization curves have similar basic trends, being valley-shaped curves with minimum values of corrosion current density (Ig). The four polarization curves all have a stable segment corresponding to a reaction platform in the anode, which is a passivation reaction area, and the corrosion current changes very little with increasing potential. Based on the analysis of the polarization curves of the four samples, the corrosion potential was obtained as shown in Table 4. Corrosion potential is the key indicator of the corrosion resistance of welds. When the corrosion potential value is lower, the corrosion resistance of the weld is better. It can be observed that with increasing single-pulse laser energy, the corrosion resistance of the welded joints with the preset nickel coating is gradually enhanced, and the corrosion resistance of Sample 4 is the best. The main reason for this is that the DP590-304 welded joints without the nickel coating are composed of dendritic lath martensite and a small amount of austenite, while the welded joints of the nickel-plated DP590-304 are composed of cellular lath martensite. Additionally, with increasing single-pulse energy, the grains in the welded joints of the nickel-plated DP590-304 become finer, so the corrosion resistance of the weld improves.

## 4. Conclusions

This paper used a preset nickel coating as the interlayer for dissimilar laser welding of DP590 steel and 304 steel. Through a comparative analysis of the microstructure, hardness, tensile strength and corrosion resistance of the welded joints, the relationship between the microstructure and the mechanical properties was systematically studied. On the basis of the results of this research, the following conclusions can be drawn:(1)The increase in the Ni element content in the weld has the benefit of inhibiting the formation of austenite and promoting the generation of lath martensite.(2)The average hardness of the DP590-304 welded joint with the preset nickel coating is increased by 12% compared with the weld without the preset nickel coating.(3)The preset nickel coating improves the tensile strength of the DP590-304 weld, and the maximum tensile strength is higher than 370 MPa.(4)The preset nickel coating in the dissimilar laser welding of DP590-304 improves the corrosion resistance of the welded joints. With increasing single-pulse energy, the cellular grains in the welding seam become denser, leading to the enhancement of the corrosion resistance of the welded joints.

## Figures and Tables

**Figure 1 materials-16-02774-f001:**
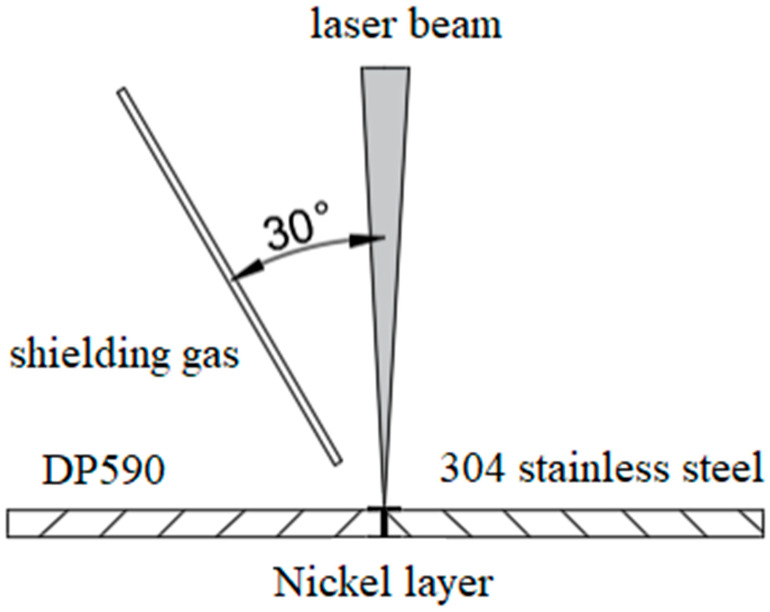
Schematic diagram of laser welding.

**Figure 2 materials-16-02774-f002:**
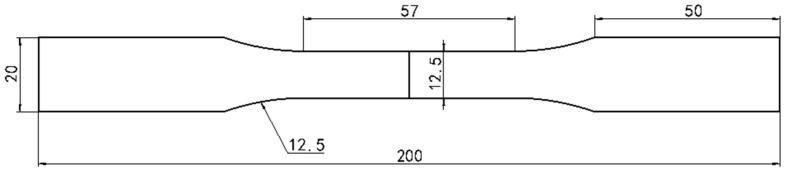
Dimensions of the tensile sample.

**Figure 3 materials-16-02774-f003:**
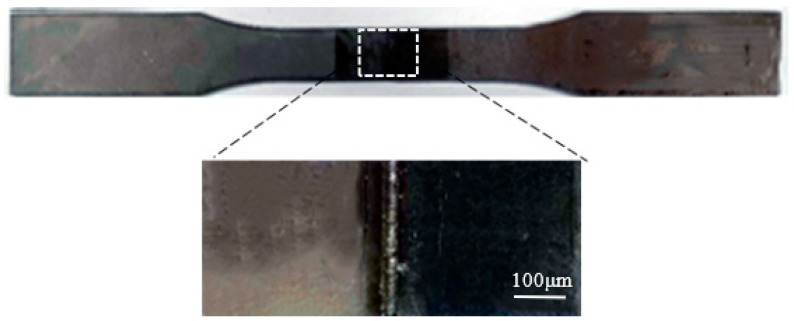
DP590-304 (preset nickel coating) tensile sample.

**Figure 4 materials-16-02774-f004:**
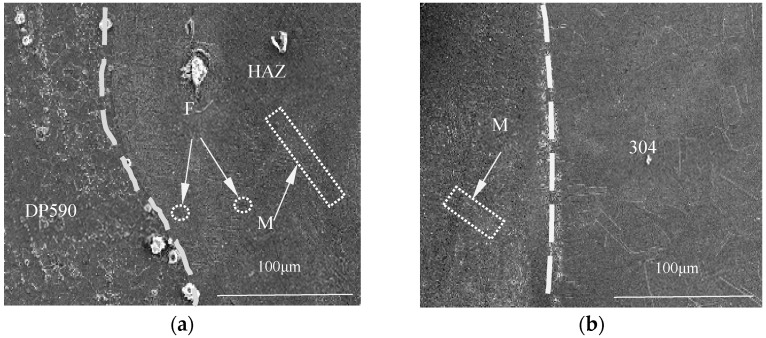
Microstructure of the boundary zones of the welded joint without the nickel coating: (**a**) near the DP590; (**b**) near the 304.

**Figure 5 materials-16-02774-f005:**
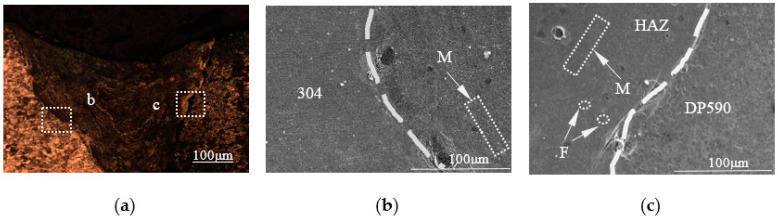
Profile and microstructures of the welded joints with the preset nickel coating at a single-pulse energy of 15 J: (**a**) profile; (**b**) boundary zone near 304; (**c**) boundary zone near DP590.

**Figure 6 materials-16-02774-f006:**
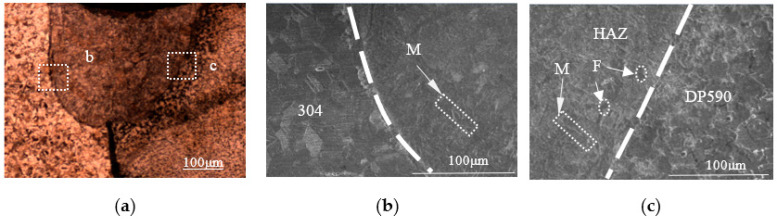
Profile and microstructures of the welded joints with the preset nickel coating at a single-pulse energy of 17 J: (**a**) profile; (**b**) boundary zone near 304; (**c**) boundary zone near DP590.

**Figure 7 materials-16-02774-f007:**
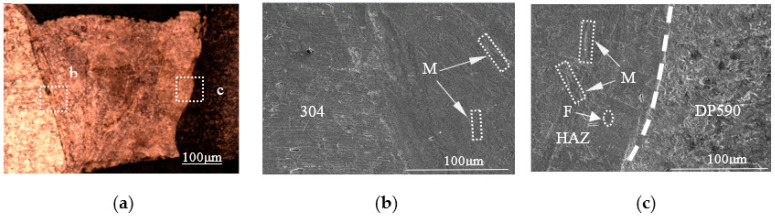
Profile and microstructures of the welded joints with the preset nickel coating at a single-pulse energy of 19 J: (**a**) profile; (**b**) boundary zone near 304; (**c**) boundary zone near DP590.

**Figure 8 materials-16-02774-f008:**
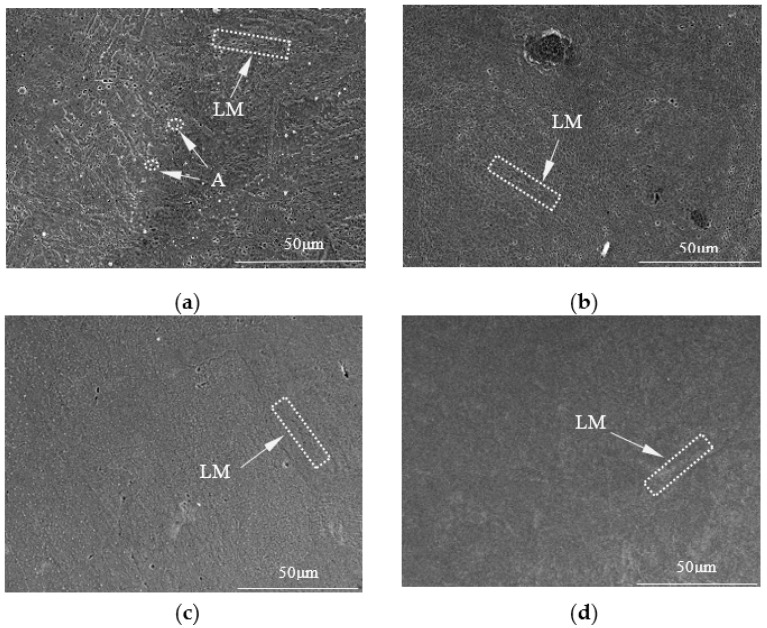
Microstructure of the DP590-304 welded joints fusion zone: (**a**) Sample 1; (**b**) Sample 2; (**c**) Sample 3; (**d**) Sample 4.

**Figure 9 materials-16-02774-f009:**
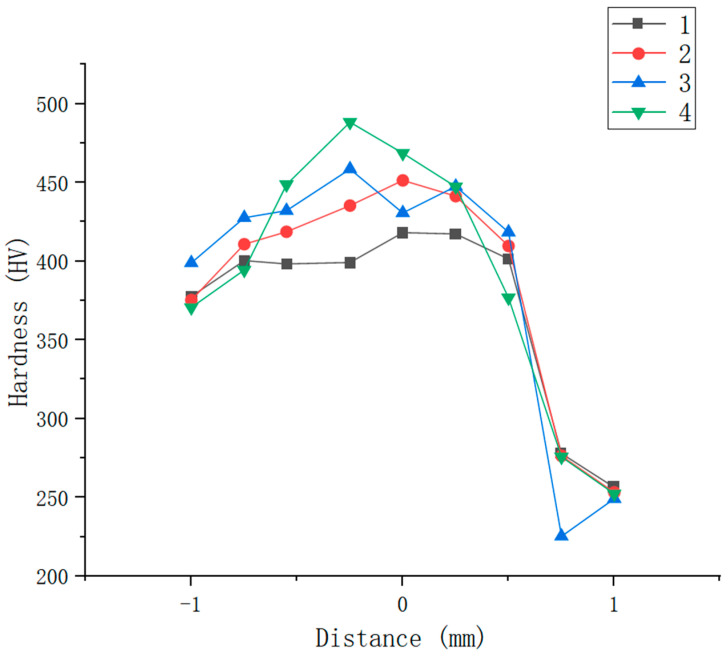
Microhardness curve of the welds of the four samples.

**Figure 10 materials-16-02774-f010:**
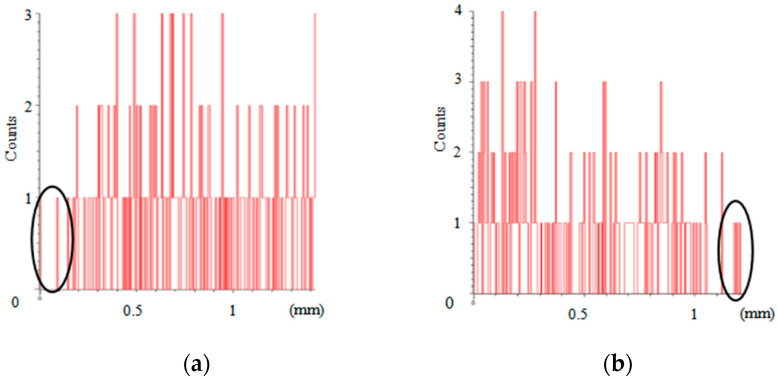
Energy spectrum analysis of the content of Ni element in the welded joints: (**a**) Sample 1; (**b**) Sample 2; (**c**) Sample 3; (**d**) Sample 4.

**Figure 11 materials-16-02774-f011:**
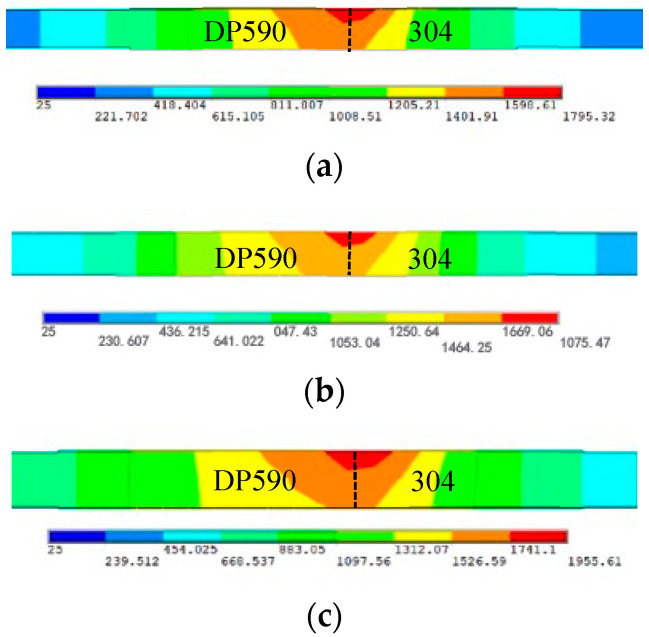
Two-dimensional temperature fields of DP590/304 weld with nickel coating under different pulse laser energies: (**a**) 15 J; (**b**) 17 J; (**c**) 19 J.

**Figure 12 materials-16-02774-f012:**
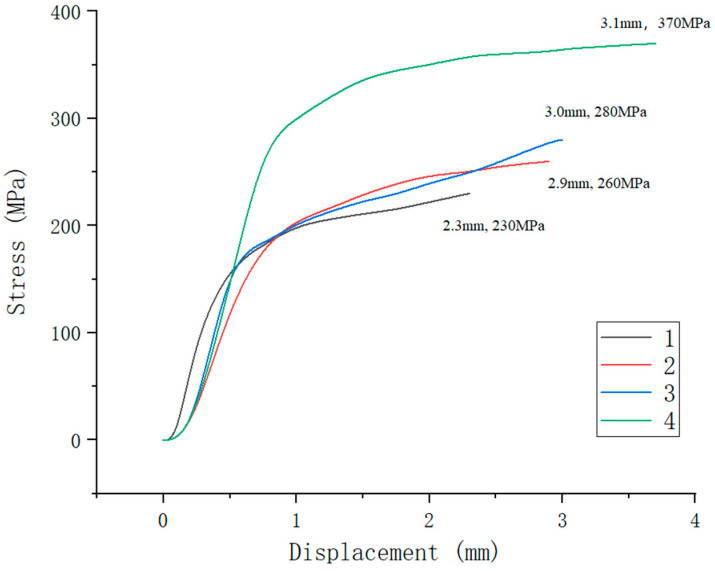
Stress–displacement curve of welded joints of the four samples.

**Figure 13 materials-16-02774-f013:**
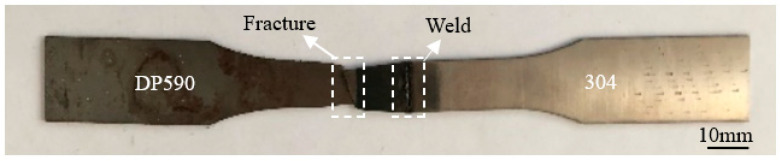
Tensile fracture Sample 4 with the fracture being on the DP590 side.

**Figure 14 materials-16-02774-f014:**
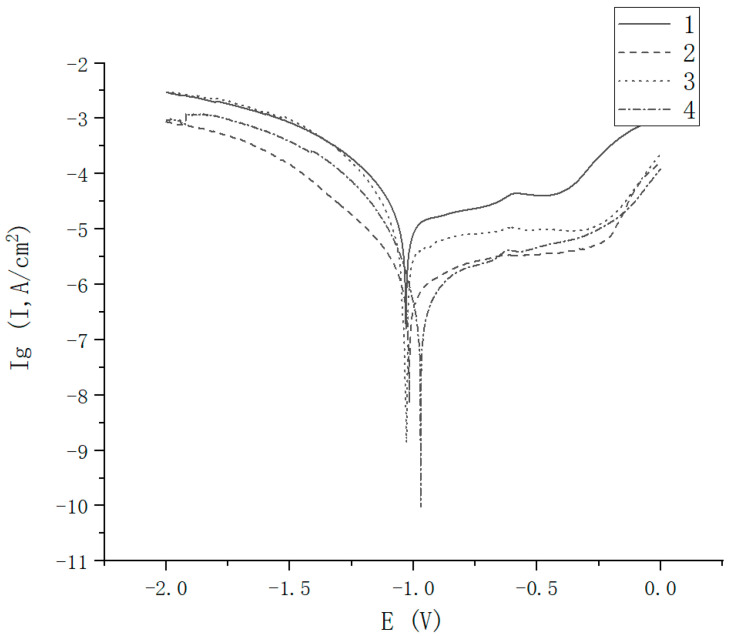
Polarization curves of the welds of the four samples.

**Table 1 materials-16-02774-t001:** Chemical composition of DP 590 and 304 stainless steel (mass score/%).

Material	C	Mn	P	Si	Ni	S	Cr	Fe
DP590	0.150	2.500	0.040	0.600	-	0.015	0.020	Bal.
304	0.060	1.950	0.028	0.300	8.930	0.030	18.27	Bal.

**Table 2 materials-16-02774-t002:** Experimental conditions of different samples.

Sample Number	Interlayer Condition	Single-Pulse Energy (J)
1	without preset nickel coating	15
2	with preset nickel coating	15
3	with preset nickel coating	17
4	with preset nickel coating	19

**Table 3 materials-16-02774-t003:** Tensile test results of the welded joints of the four samples.

Sample Number	YS (MPa)	UTS (MPa)	Elongation (%)
1	160	230	1.15
2	130	260	1.45
3	155	280	1.5
4	165	370	1.55

**Table 4 materials-16-02774-t004:** Polarization curve analysis of the welds in the four samples.

Sample Number	Corrosion Potential (V)
1	−1.0315
2	−1.0192
3	−1.0075
4	−0.971

## Data Availability

Not applicable.

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
