# Peer review of "Microstructure and Mechanical Properties of Laser-Welded Joints between DP590 Dual-Phase Steel and 304 Stainless Steel with Preset Nickel Coating"

_materials, 2023, doi:10.3390/ma16072774_

Round 1
Reviewer 1 Report
Dear Authors,
The reviewed manuscript titled: “Microstructure and Mechanical Properties of Laser-Welded Joints Between DP590 Dual-Phase Steel and 304 Stainless Steel with Preset Nickel Coating” describes an experimental study aimed at assessing the weldability issues of laser welded austenitic stainless and dual phase automotive steel grades.
The article deals with a current and relevant issue, it is well prepared. The subject is current and important, and the content of the manuscript made a good overall impression on me. The work is in line with the current trend of the increasing number of articles on the possibility of making dissimilar joints, and the basic novelty is the original set of welded materials, quite rarely found in the literature. But I have a few comments, which I present below:
The abstract and Materials and methods chapter lack information that austenitic steel was welded.
I propose to remove the sentence: "The DP590 14 dual phase steel and 304 stainless steel both are 1 mm thickness." and information about the thickness of the welded elements should be added to the previous sentence.
I also propose to make the following changes in the keywords section:
laser welding; dissimilar joints; austenitic steel; dual phase steel; nickel coating; microstructure; mechanical properties
Materials and methods:
Please complete the information about the welding procedure so that the reader can repeat it based on the description: please add information about the purity of the shielding gas. Was backing gas used?
Please add the names and addresses of the manufacturers of all devices used in accordance with the MDPI guidelines.
Figure 9: please mark where is the weld. I think the figure is too small. Figure caption is not correct. The font is not clearly visible.
Figure 11: Figure caption is not correct. The font is not clearly visible.
Table 4 caption: "Polarization curve analysis of the weld joints." This form suggests that tests were carried out in all zones of welded joints, and the content of the work shows that only weld were tested.
Conclusions:
The first conclusion doesn't sound right. "Improvement" does not show the direction of change. Maybe it would be better: „Increase”?
Terminology: please replace: "weld joint" with: "welded joint"
Metallographic terms do not need to be capitalized, e.g. Lath Martensite, Austenite
Please replace: "Martensites" and "Ferrites" with: "martensite" and "ferrite"
Please add spaces before the units.
Finally, please consider using the information contained in the articles: https://doi.org/10.3390/ma16041334, https://doi.org/10.3311/PPme.21007 and works of Dr. Chandan Pandey who also does laser welding of dissimilar joints.
Author Response
Dear Reviewer,
I am pleased to note the favorable comments in your opening sentence. I have read the comments and recommendation carefully. Based on the former research, I spent about three weeks on the revision of the paper.
Now, I will answer your question and illustrate the revision.
1. I propose to remove the sentence: "The DP590 14 dual phase steel and 304 stainless steel both are 1 mm thickness." and information about the thickness of the welded elements should be added to the previous sentence.
Answer: I have removed the sentence and added the information in the previous sentence.
The sentence of my revision: To overcome such brittleness, adding preset coating on the base materials as being interlayer is a method for attaining reliable welded joints. Nd:YAG laser butt welding of DP590 dual phase steel and 304 stainless both with a thickness of 1 mm has been performed with preset nickel coating as an interlayer using electroplating process.
2. I also propose to make the following changes in the keywords section:
laser welding; dissimilar joints; austenitic steel; dual phase steel; nickel coating; microstructure; mechanical properties
Answer: As you pointed out, my original paper needs to add new keywords.
Keywords in my revision: laser welding; dissimilar joints; austenitic steel; dual phase steel; nickel coating; microstructure; mechanical properties
3. Materials and methods:
3.1 Please complete the information about the welding procedure so that the reader can repeat it based on the description: please add information about the purity of the shielding gas. Was backing gas used?
Answer: As you pointed out, my original paper is lacks a description of auxiliary gases for laser welding.
In my revision: The shielding gas is argon with the purity of 99.9% and the flow rate of 25 L/min.
3.2 Please add the names and addresses of the manufacturers of all devices used in accordance with the MDPI guidelines.
Answer: As you pointed out, my original paper is lacks a description of all devices.
In my revision, I have added the descriptions of all used devices.
A Nd:YAG laser welding machine (JHM-1GXY-500, CHUTIAN, China);
A microhardness tester (TMVS-1, Beijing Times, China);
An electrochemical workstation (CHI666C, Shanghai Chenhua, China);
A confocal microscope (Axio Scope A1, ZEISS, Germany);
A scanning electron microscope (S-3400N, Hitachi, Japan);
An electronic universal tensile testing machine (CMT5105, MTS, USA).
3.3 Figure 9: please mark where is the weld. I think the figure is too small. Figure caption is not correct. The font is not clearly visible.
3.4 Figure 11: Figure caption is not correct. The font is not clearly visible.
Answer: I have changed the Figure 9 and Figure 11.
3.5 Table 4 caption: "Polarization curve analysis of the weld joints." This form suggests that tests were carried out in all zones of welded joints, and the content of the work shows that only weld were tested.
Answer: I have changed the title of Table 4:
Table 4. Polarization curve analysis of the welds from four Samples
4. Conclusions:
The first conclusion doesn't sound right. "Improvement" does not show the direction of change. Maybe it would be better: „Increase”?
Answer: As you pointed out, “Improvement” is not used properly in my original paper’s conclusion.
In first conclusion of my revision: The increase of Ni element content in the weld has benefit on in-hibiting the formation of austenite and promoting the generation of lath martensite.
5. Terminology: please replace: "weld joint" with: "welded joint"
Answer: all ‘weld joint’ have changed to be ‘welded joint’ in my revision.
6. Metallographic terms do not need to be capitalized, e.g. Lath Martensite, Austenite
Please replace: "Martensites" and "Ferrites" with: "martensite" and "ferrite"
Answer: As you pointed out, all metallographic terms do not need to be capitalized. I have changed all metallographic terms with ‘martensite’, ‘ferrite’ and ‘austenite’.
7. Please add spaces before the units.
Answer: I have added spaces before all units in my revision.
8. Finally, please consider using the information contained in the articles: https://doi.org/10.3390/ma16041334, https://doi.org/10.3311/PPme.21007 and works of Dr. Chandan Pandey who also does laser welding of dissimilar joints.
Answer: Thank you for the informative reminder of the references. I have added the research content of article of https://doi.org/10.3311/PPme.21007 in the Introduction of my paper.
In the introduction of my revision:
Varbai et al. [15] used gas tungsten arc welding to obtain the joint between lean duplex stainless steel and 304 austenitic stainless steels. Their investigations reveal that the best pitting corrosion resistance is achieved using the 308L welding rod with argon shielding gas, where the weight loss was 1.6% after the 24 hours immersion test.
In the References of my revision:
- Varbai, B.; Bolyhos, P.; Kemény, D.M.; Májlinger K. Microstructure and corrosion properties of austenitic and duplex stainless steel dissimilar joints. Period. Polytech-mech. 2022, 66, 344-349.
All in all, I greatly appreciate your help. Thank you very much.
Hua Zhang
21-03-2023

Reviewer 2 Report
The manuscript is related Microstructure and Mechanical Properties of Laser-Welded Joints Between DP590 Dual-Phase Steel and 304 Stainless Steel with Preset Nickel Coating. However, before publication some correction is needed.
1. The introduction is general and needs to be supplemented with new references related to dissimilar materials welding. Authors can use the following references to complete their work: “Microstructure and mechanical properties of ultrasonic spot welding TiNi/Ti6Al4V dissimilar materials using pure Al coating”, “Microstructure and mechanical properties of dissimilar NiTi/Ti6Al4V joints via back-heating assisted friction stir welding”, “Investigation of welding crack in micro laser welded NiTiNb shape memory alloy and Ti6Al4V alloy dissimilar metals joints”, “Effect of welding thermal treatment on the microstructure and mechanical properties of nickel-based superalloy fabricated by selective laser melting”
2. The authors should analyze and discuss the heat input. Authors can use the following references to complete their work: “Effect of heat input on interfacial characterization of the butter joint of hot-rolling CP-Ti/Q235 bimetallic sheets by Laser + CMT”
3. The name, model and company related to the used equipment should be mentioned in the materials and methods section. Welding machine, SEM, OM, …
4. Extract the data related to figure 11 and present it as a table in the manuscript, YS, UTS, Elongation, …
5. The discussion about the presence of nickel and its effect on welding should be further discussed. Authors can refer to the following articles. “Isolated Ni atoms induced edge stabilities and equilibrium shapes of CVD-prepared hexagonal boron nitride on Ni(111) surface”, “Microstructural origin and control mechanism of the mixed grain structure in Ni-based superalloys”
6. The recent references should be used to complete the discussion of the corrosion section and the discussion should be referenced. Authors can use the following article to complete the discussion: “Hydrogen embrittlement behavior of SUS301L-MT stainless steel laser-arc hybrid welded joint localized zones”
7. There should be a space between the numbers and their units
8. Line 84, 85: Subtitle the numbers
9. Add scale bar for Figure 3
10. Add error bar figure 9
11. Add Y axis title in figure 10
Author Response
Dear Reviewer
I am pleased to note the favorable comments in your opening sentence. I have read the comments and recommendation carefully. Based on the former research, I spent about three weeks on the revision of the paper.
Now, I will answer your question and illustrate the revision.
1. The introduction is general and needs to be supplemented with new references related to dissimilar materials welding. Authors can use the following references to complete their work: “Microstructure and mechanical properties of ultrasonic spot welding TiNi/Ti6Al4V dissimilar materials using pure Al coating”, “Microstructure and mechanical properties of dissimilar NiTi/Ti6Al4V joints via back-heating assisted friction stir welding”, “Investigation of welding crack in micro laser welded NiTiNb shape memory alloy and Ti6Al4V alloy dissimilar metals joints”, “Effect of welding thermal treatment on the microstructure and mechanical properties of nickel-based superalloy fabricated by selective laser melting”
Answer: As you pointed out, my original paper needs to be supplemented with some new references related to dissimilar mataerials welding. Thank your help for giving some references.
I have added three references in the introduction of my revision:
Xie et al. [14] had investigated microstructure and mechanical properties of ultrasonic spot welding TiNi/Ti6Al4V dissimilar materials using pure Al coating. The study re-veals that TiNi/Ti6Al4V dissimilar materials are successfully joined and the tensile shear load reached the maximum value of 931 N.
In addition, Huaibo Deng et al. [24] had compared back-heating assisted friction stir welding with conventional friction stir welding to confirm the feasibility of dissimilar NiTi/Ti6Al4V joints fabricated via friction stir welding. They found that the defect-free joint was obtained when preheating temperature at 200 ℃ during back-heating as-sisted friction stir welding. Chen Yuhua et al. [25] had investigated the welding crack in micro laser welded NiTiNb shape memory alloy and Ti6Al4V alloy dissimilar metals joints. The study reveals that brittle Ti2Ni phase with more contents is the main reason of crack formation along with large stress concentration.
In the References of my revision:
- Xie, J.L.; Chen, Y.H.; Yin L.M.; Zhang, T.M.; Wang, S.L.; Wang, L,T. Microstructure and mechanical properties of ultrasonic spot welding TiNi/Ti6Al4V dissimilar materials using pure Al coating. J. Manuf. Processes. 2021, 64, 473–
- Deng, H.B.; Chen, Y.H.; Jia, Y.L.; Pang, Y.; Zhang, T.M.; Wang, S.L.; Yin, L.M. Microstructure and mechanical properties of dissimilar NiTi/Ti6Al4V joints via back-heating assisted friction stir welding. J. Manuf. Processes. 2021, 64, 379-391.
- Chen, Y.H.; Mao, Y.Q.; Lu, W.W.; He, P. Investigation of welding crack in micro laser welded NiTiNb shape memory alloy and Ti6Al4V alloy dissimilar metals joints. Opt. Laser Technol. 2017, 91,197-202.
2. The authors should analyze and discuss the heat input. Authors can use the following references to complete their work: “Effect of heat input on interfacial characterization of the butter joint of hot-rolling CP-Ti/Q235 bimetallic sheets by Laser + CMT”
Answer: Thank your help for giving the reference about heat input of laser.
In 3.2 of my revision, I have added a figure about two-dimensional temperature fields of DP590/304 weld with Nickel coating under different pulse laser energy:(a) 15 J; (b) 17 J; (c) 19 J. Combined with this figure, the distribution characteristics of nickel elements in the molten pool and the changes of weld hardness were were further analyzed.
- The name, model and company related to the used equipment should be mentioned in the materials and methods section. Welding machine, SEM, OM, …
Answer: As you pointed out, my original paper is lacks a description of all devices.
In my revision, I have added the descriptions of all used devices.
A Nd:YAG laser welding machine (JHM-1GXY-500, CHUTIAN, China);
A microhardness tester (TMVS-1, Beijing Times, China);
An electrochemical workstation (CHI666C, Shanghai Chenhua, China);
A confocal microscope (Axio Scope A1, ZEISS, Germany);
A scanning electron microscope (S-3400N, Hitachi, Japan);
An electronic universal tensile testing machine (CMT5105, MTS, USA).
- Extract the data related to figure 11 and present it as a table in the manuscript, YS, UTS, Elongation, …
Answer: In my revision, I have added Table 3 as following:
Table 3. Tensile test data results of welded joints from four Samples
|
Sample Number |
YS (MPa) |
UTS(MPa) |
Elongation (%) |
|
1 |
160 |
230 |
1.15 |
|
2 |
130 |
260 |
1.45 |
|
3 |
155 |
280 |
1.5 |
|
4 |
165 |
370 |
1.55 |
- The discussion about the presence of nickel and its effect on welding should be further discussed. Authors can refer to the following articles. “Isolated Ni atoms induced edge stabilities and equilibrium shapes of CVD-prepared hexagonal boron nitride on Ni(111) surface”, “Microstructural origin and control mechanism of the mixed grain structure in Ni-based superalloys”
Answer: Thank your help for give some reference. However, the topic of first reference is to CVD-prepared hexagonal boron nitride and the topic of second reference is to Preparation of Ni-based superalloys. The topics of these two references lack similarity to the topic of dissimilar material welding in my paper. I have added some discussion about the presence of nickel in 3.2 Microhardness of my revision.
- The recent references should be used to complete the discussion of the corrosion section and the discussion should be referenced. Authors can use the following article to complete the discussion: “Hydrogen embrittlement behavior of SUS301L-MT stainless steel laser-arc hybrid welded joint localized zones”
Answer: Thank your help for give a reference about corrosion section. The topic of the reference is hydrogen embrittlement behavior of welded joint. Electrochemical hydrogen charging process at room temperature was employed on the polished samples in 0.5 mol/L H2SO4 + 2 g/L NH4SCN solution under a current density of 10 mA/cm2 during slow strain rate tensile (SSRT) tests with a strain rate of 10−5 s-1, where the tensile samples worked as cathode and a platinum sheet worked as anode.
However, There is difference between the reference and my paper. In my paper, the corrosion resistance of welded joint was tested with an electrochemical workstation. The electrochemical workstation has a three-electrode system, the working electrode is the welded joint, the reference electrode is a saturated calomel electrode and the auxiliary electrode is a platinum electrode. The working condition of the three-electrode system is 0.5% NaCl solution. In my revision, the corrosion resistance test results have been compared and analyzed.
- There should be a space between the numbers and their units
Answer: I have added spaces before all units in my revision.
8. Line 84, 85: Subtitle the numbers
9. Add scale bar for Figure 3
10. Add error bar figure 9
11. Add Y axis title in figure 10
8-11 Answer: I have made the corresponding modification in my revision.
All in all, I greatly appreciate your help. Thank you very much.
Hua Zhang
21-03-2023

Reviewer 3 Report
The development of a laser welding process for DP590 dual-phase steel and 304 stainless steel with present nickel coating is a laborious task, as it requires a lot of experimentation. After all, the thermal cycle of laser welding is characterized by ultra-high heating and cooling rates of the weld metal and the heat-affected zone, high temperature gradients in the heat-affected zone (HAZ), as well as short exposure times. This affects the ongoing phase and structural changes in the HAZ and, as a result, determines the mechanical properties of the welded joint, which requires more detailed research in this area. Therefore, the results of the article are important for science and practice. There are a few suggestions for improving the article:
1. I propose to correct the captions. If there are designations in the figure, they should be in the caption. The figure should be clear, without looking for additional explanations in the text of the article.
2. The authors in fig. 4-8 describe austenite(A) and transformation of the weld structure into lath martensite. I unfortunately do not see these microstructures. If the authors see them, please indicate the sizes of grains or other structural elements. In my opinion, metallography at such small magnifications is impossible, except for macroanalysis.
3. Figure 10 is located on 0.5 pages, and it description was limited to two lines. I think this description is not enough to understand this figure. What is shown on it? What values are plotted along the axes. It is desirable to clarify this.
4. It is necessary to clarify how the Fig.11 was obtained. Where measurements of displacement were taken. What was the measurement base (distance between the sensor arms)? Was it only the weld zone that deformed, or did the adjacent metal zone also deform?
5. I suggest in fig. 9 schematically indicate the main zones of the weld.
6. I invite the authors to study the article https://link.springer.com/article/10.1007/s11003-019-00250-x Perhaps it will be useful to the authors in clarifying the description of their results.
Author Response
Dear Reviewer,
I am pleased to note the favorable comments in your opening sentence. I have read the comments and recommendation carefully. Based on the former research, I spent about three weeks on the revision of the paper.
Now, I will answer your question and illustrate the revision.
- I propose to correct the captions. If there are designations in the figure, they should be in the caption. The figure should be clear, without looking for additional explanations in the text of the article.
Answer: The titles of the figures in my revision have been corrected with full informations.
- The authors in fig. 4-8 describe austenite(A) and transformation of the weld structure into lath martensite. I unfortunately do not see these microstructures. If the authors see them, please indicate the sizes of grains or other structural elements. In my opinion, metallography at such small magnifications is impossible, except for macroanalysis.
Answer: As you pointed out, the microstructures in Fig.4-8 need to be further analzed. In my revision, I have added some sentences to describe the results about microstructures:
From qualitative analysis, the basis for judging of austenite (A) is the white dot in and the basis for judging of lath martensite (LM) is the narrow profile. Comparing the three SEM pictures for qualitative analysis, it can be seen that the structures of the dissimilar welds under different single laser pulse energy have the same characteristic of being less austenites and more lath martensites.
I have to admit that there is a qualitative analysis and can not be precise to the sizes of grains.
- Figure 10 is located on 0.5 pages, and it description was limited to two lines. I think this description is not enough to understand this figure. What is shown on it? What values are plotted along the axes. It is desirable to clarify this.
Answer: As you pointed out, the description about Figure 10 is not enough.
In my revision, I have added unit of the ordinate in Figure 10. I have added a figure about the two-dimensional temperature fields of DP590/304 weld with Nickel coating under different pulse laser energy. Combined Figure 9-11, the relationship between microhardness and the distribution of nickel element was further analyzed.
- It is necessary to clarify how the Fig.11 was obtained. Where measurements of displacement were taken. What was the measurement base (distance between the sensor arms)? Was it only the weld zone that deformed, or did the adjacent metal zone also deform?
Answer: The datas of Fig.11 were from the tensile tests of welded samples with an electronic universal tensile testing machine (CMT5105, MTS, USA) . The tensile tests were performed in standard mode. I have added a figure of tensile fracture sample 4 with the fracture being on the side of DP590. From the added figure, The fracture zone is on the side of DP590 base metal and the distance between the fracture zone and the weld zone is about 8mm. It can be seen that the deformation area in the tension of the welded sample is concentrated on the side of DP590 and there is no obvious deformation in the weld zone and 304 base metal. It also confirms that the tensile strength of the weld of Sample 4 is better than that of the base metal DP590.
- I suggest in fig. 9 schematically indicate the main zones of the weld.
Answer: In fig.9, I have added a labe with two dotted lines to indicate the main zones of the weld in my revision.
- I invite the authors to study the article https://link.springer.com/article/10.1007/s11003-019-00250-x Perhaps it will be useful to the authors in clarifying the description of their results.
Answer: Thank your help for giving an important reference. The quality of CO2 laser-welded bodywork steel sheets was analyzed in the reference. It provides an important reference for my follow-up research.
All in all, I greatly appreciate your help. Thank you very much.
Hua Zhang
21-03-2023

Round 2
Reviewer 2 Report
Accept in present form
Reviewer 3 Report
Accept.